# Unmanned Aerial Traffic Management System Architecture for U-Space In-Flight Services

**Carlos Capitán \*** , **Héctor Pérez-León, Jesús Capitán** , **Ángel Castaño** and **Aníbal Ollero**

Engineering School, University of Seville, Av. de los Descubrimientos, s/n, 41092 Seville, Spain; hectorperez@us.es (H.P.-L.); jcapitan@us.es (J.C.); castano@us.es (Á.C.); aollero@us.es (A.O.)
\* Correspondence: ccapitan@us.es

**Featured Application: This work can be applied to integrate autonomous unmanned aerial vehicles in civil airspace.**

**Abstract:** This paper presents a software architecture for *Unmanned aerial system Traffic Management* (UTM). The work is framed within the U-space ecosystem, which is the European initiative for UTM in the civil airspace. We propose a system that focuses on providing the required services for automated decision-making during real-time threat management and conflict resolution, which is the main gap in current UTM solutions. Nonetheless, our software architecture follows an open-source design that is modular and flexible enough to accommodate additional U-space services in future developments. In its current implementation, our UTM solution is capable of tracking the aerial operations and monitoring the airspace in real time, in order to perform in-flight emergency management and tactical deconfliction. We show experimental results in order to demonstrate the UTM system working in a realistic simulation setup. For that, we performed our tests with the UTM system and the operators of the aerial aircraft located at remote locations with the consequent communication issues, and we showcased that the system was capable of managing in real time the conflicting events in two different use cases.

**Keywords:** UTM; system architecture; U-space; UAS





## 1. Introduction

In the last few years, there has been a clear trend to use *Unmanned Aircraft Systems* (UAS), or drones, for many commercial and civil applications. There are studies [1] estimating that up to 400,000 drones will be providing services in the airspace by 2050, with a total market value of 10 billion euros per year by 2035. Last-mile delivery [2], surveillance [3], infrastructure inspection [4], traffic monitoring [5], media production [6], or managing health emergency situations [7] are just a few examples of the wide spectrum of drone applications. Indeed, the integration of UAS in the civil airspace is probably one of the most revolutionary events for *Air Traffic Management* (ATM) since the beginning of its implementation. Although ATM has been traditionally based on voice communication through an *Air Traffic Control* (ATC) entity, its bounded workload and communication capacities turn this centralized resource into a bottleneck for system scalability. Therefore, the rise of UAS operations brings the need for a new paradigm for airspace management, where digital communication will play a key role, and the responsibilities will be shared among different stakeholders instead of a single central actor.

There are already some initiatives to integrate UAS into civil airspace and fulfill their operational requirements [8]. The *National Aeronautics and Space Administration* (NASA) has created the concept for *UAS Traffic Management* (UTM) [9] to enable safe, large-scale operations with UAS in low-altitude airspace [10]; whereas Europe has extended this UTM concept by proposing the *U-space* ecosystem [11]. More specifically, an overview of the

U-space ecosystem recently proposed by the *European Aviation Safety Agency* (EASA) [12] is depicted in Figure 1. The idea is to have a *U-space service Provider Platform*, which is a server running on the cloud, as the core component. There, the UTM system consists of a software architecture that provides U-space services to the different actors in the U-space ecosystem using as a bridge the *U-space Service Manager* (USM), which is a specific module of this UTM system.

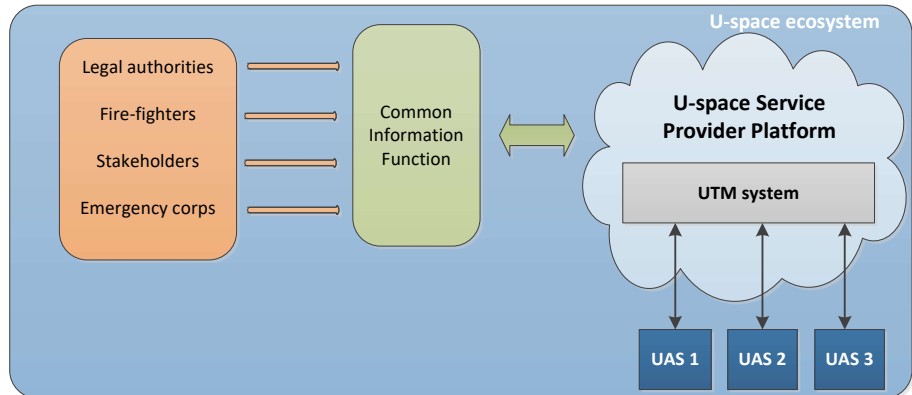

**Figure 1.** Overview of the U-space ecosystem proposed by EASA [12]. The UTM system offers U-space services to the different actors and runs on a remote server called U-space Service Provider Platform.

Currently, the community is in the process of further developing these U-space services. In this paper, we take a first step and propose a novel software architecture that aims to serve as a common framework for implementing and integrating U-space services. Our solution is being developed within the context of the European project GAUSS (https://projectgauss.eu, accessed on 26 April 2021), whose main objective is leveraging high-performance positioning functionalities provided by the Galileo ecosystem for U-space operations, including a validation phase with actual fixed-wing and rotary-wing UAS (see Figure 2). We present an architecture that is service-oriented and safety-centered, and that allows the airspace actors to abstract from specific UAS technologies. Besides, we implement a set of U-space services to manage complete UAS operations, but focusing on *in-flight* services (i.e., those required to handle the operations during the flight phase). Nonetheless, the architecture is modular and flexible enough to be extended with additional functionalities as new services become functional.

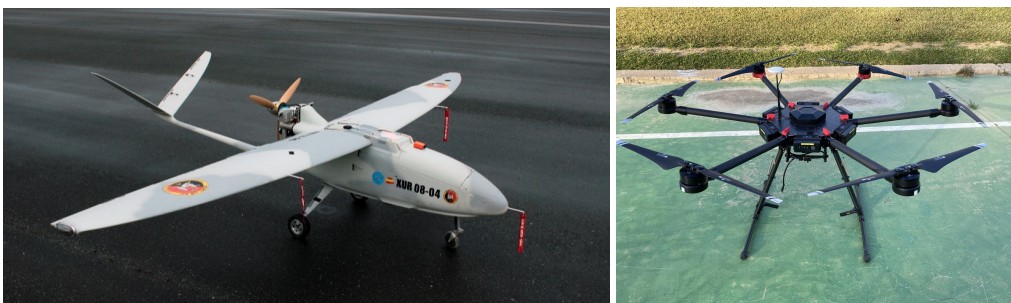

**Figure 2.** The Atlantic I (**left**) and DJI M600 (**right**) UAS will be used to validate the UTM functionalities developed in the GAUSS project.

Our main contributions are as follows. First, we introduce the main concepts and the roadmap for the U-space initiative, and we review other relevant works about UTM (Section 2). Second, we analyze the design properties for our UTM architecture (Section 3). Given a series of desired architectural guidelines (Section 3.1), we propose the open-source *Robot Operating System* (http://www.ros.org, accessed on 26 April 2021) as underlying middleware for our UTM system (Section 3.2). Third, we contribute with a new UTM system architecture implementing the U-space concept (Section 4). Our proposal represents

a general framework for U-space services, which is modular, flexible, and technology-agnostic; but we describe our specific implementation for a set of core in-flight services dealing with unexpected UAS conflicts during their flight phase. Our software framework integrates automated decision-making procedures, which is one of the main gaps for current UTM solutions. Additionally, we show an actual realization of our UTM architecture that is available as open-source software for the community, and we demonstrate its capabilities (Section 5). In order to showcase the correct integration of all our components and services, we have defined use cases for UAS operations involving all the developed functionalities (Section 5.1); and we have assessed our results in terms of performance by running the whole system in a realistic simulation setup for multi-UAS operations (Sections 5.2 and 5.3). Finally, we draw the main conclusions of this work and point at future lines for further development (Section 6).

## 2. Background

In this section, we introduce the U-space initiative and its offered services, as well as its development roadmap. Then, we review the related work about UTM systems.

### 2.1. U-Space

U-space is a collaborative effort among researchers, industry, and regulators to enable the integration of UAS operations within the civil airspace, providing UAS situational awareness and digital communication with manned aviation, the ATM service providers, and the legal authorities. There exists a roadmap [11] to deploy U-space in Europe, consisting of the four phases depicted in Figure 3. Each phase will propose a new set of services with increasing complexity and integration level between UAS and manned aircraft, as well as an upgraded version of existing services in the previous phases.

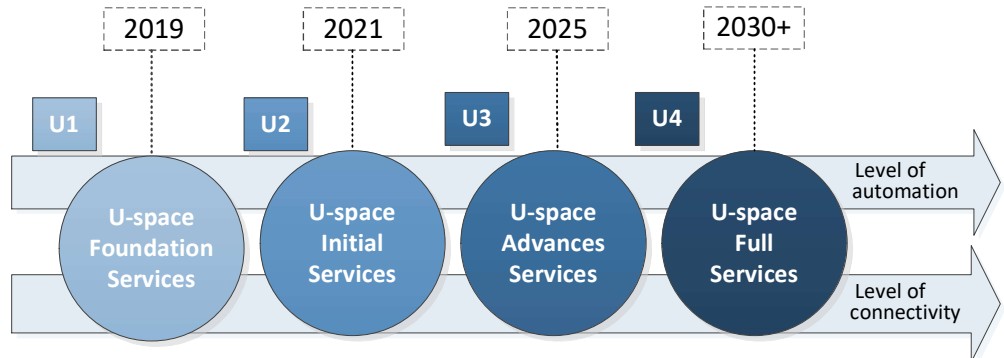

**Figure 3.** The implementation roadmap for the U-space initiative [13], consisting of 4 deployment phases.

The detailed functional system architecture is still under development, but there is already a list of services defined for each deployment phase [14], and a report with the current progress of their implementation and deployment [15]. Table 1 depicts these services and their current level of implementation in Europe.

**Table 1.** The U-space services for each development phase, together with their current implementation level. Our system focuses on in-flight services to handle unexpected events during the flight phase of the UAS.

| Phase | Service | Overall Implementation Level | Covered in Our UTM System |
|---|---|---|---|
| U1 Foundation Services | E-registration | 19% | |
| | E-identification | 17% | |
| | Pre-tactical geofencing | 23% | |
| U2 Initial Services | Tactical geofencing | 13% | |
| | Flight planning management | 6% | |
| | Weather information | 3% | |
| | Tracking | 4% | ✓ |
| | Monitoring | 5% | ✓ |
| | Drone aeronautical information management | 18% | |
| | Procedural interface with ATC | 20% | |
| | Emergency management | 9% | ✓ |
| | Strategic deconfliction | 6% | |
| U3 Advanced Services | Dynamic geofencing | 5% | |
| | Collaborative interface with ATC | 8% | |
| | Tactical deconfliction | 0% | ✓ |
| | Dynamic capacity management | 4% | |
| U4 Full Services | To be defined | 0% | |

The U-space framework proposes a UTM system as the software architecture that provides services to the different U-space actors. A possible classification for the services is depending on whether they are activated in the UAS pre-flight phase or during the flight:

- Pre-flight services are those related with the functionalities needed to prepare and schedule a UAS operation. The vehicle and the operator need to register (*E-registration*), and the initial flight plan has to be handled before being accepted (*Flight planning management*). Then, the pilot may get assistance through information about predefined restricted areas (*Pre-tactical geofencing*) and the resolution of possible conflicts before flying (*Strategic deconfliction*).
- In-flight services deal with the functionalities required to handle the operation after the UAS flight has started. This means the possibility to update the operator (*Tactical geofencing*) or the UAS itself (*Dynamic geofencing*) with geofencing information during the flight, and to track the current position and predicted trajectory for each UAS (*Tracking*). This updated information is then used to create a situation of the airspace (*Monitoring*) and to generate warnings and contingency actions under possible threats (*Emergency management*). Alternative plans could also be suggested in-flight to maintain the required separation between aircraft and with geofences (*Tactical deconfliction*).
- There are other services that could be used either before flying or during the flight. These are functionalities that aim to provide identification (*E-identification*), weather forecasts (*Weather Information*), or more generic information (*Drone Aeronautical Information Management*), to create an interface with the ATC (*Procedural Interface with ATC* and *Collaborative interface with ATC*), and to control and manage the UAS density in the airspace (*Dynamic Capacity Management*).

According to Table 1 and to our study of the state of the art, in-flight services have been less addressed by UTM systems in general, with a notorious integration gap still

existing. In this paper, we focus on in-flight functionalities to develop a UTM system, although our architecture is general enough to cover all kinds of services. In particular, we integrate those services related to the management of unexpected events while the UAS are flying, namely tracking, monitoring, emergency management, and tactical deconfliction. These services belong to the U2 and U3 implementation phases, which are scheduled to be developed between 2021 and 2029.

### 2.2. Related Work

The development of completely operational UTM systems is still at an early stage, even though it has recently become a growing field. The authors in [16] define what a UTM system is, and they give an overview of both a physical UTM architecture and a UTM software manager based on automated services. Big companies are one of the major parties interested in boosting the deployment of UTM. For instance, Google has proposed an ecosystem [17] where all UAS should be equipped with communication and *sense & avoid* technologies in order to perform cooperative flights when encountering other UAS or manned aircraft. In their proposal, the separation and planning services would be provided by an *Airspace Service Provider*. Furthermore, Amazon has put forward a one-operator-to-many-vehicle model [18], where the decision-making authority gets significantly distributed among the operators.

Additionally, there exist several commercial UTM system applications in the market. They implement most pre-flight services, but just partially a few in-flight services. For instance, Airmap [19] has its focus on UAS registration, geographic information systems, flight communication, traffic monitoring, and user interfaces. The Unifly platform [20] connects authorities with pilots to safely integrate UAS into the airspace. On the one hand, the authorities can visualize and approve the UAS flights, as well as manage *No Flight Zones* in real time. On the other hand, the pilots can manage their UAS (e.g., with the E-registration, E-identification, and Flight plan management services) and they can plan and receive flight approvals aligned with international and local regulations. Another framework is the Thales ECOsystem UTM [21], which integrates UAS and pilot registration. ECOsystem provides a flight planning functionality, using airspace rules and situational awareness as guidelines. It also includes tools to manage map overlays and 3D terrain views.

The aforementioned UTM applications offer pre-flight UTM services and some in-flight capabilities such as UAS tracking. Even though they are capable of publishing real-time information about the UAS, they do not manage operations autonomously during the flight phase. Moreover, it is important to highlight that all those applications are commercial products that are not available for the community as open software.

The scientific community has also been putting effort into functional UTM frameworks; a recent review of related works can be seen in [22]. A prototype UTM for flight surveillance has recently been proposed in Taiwan [23]. One of its core properties is the capability to monitor vehicles, being the ADS-B (*Automatic Dependent Surveillance Broadcast*) technology used for surveillance. There is a pre-flight procedure to schedule and approve flights, and then the UTM system can send surveillance alerts during the operation, though all the decisions for conflict resolution are up to the pilot. Another UTM system has been presented in Sweden [24]. It incorporates a complete toolkit to manage traffic, geofences, flight altitude segregation as in the general aviation, and complex visualization. This research has also identified problems that dense traffic in the low-level airspace will bring to the city users, by simulating the future urban airspace. In general, the functionalities of the aforementioned systems have only been demonstrated through simplistic simulations, and quite a few works have been devoted to field flight campaigns for preliminary tests [25,26]. We have also proposed in a previous work [27] a more realistic simulator for UAS operations based on the ROS middleware and the 3D simulation suite Gazebo (http://gazebosim.org, accessed on 26 April 2021). In that work, we introduced a preliminary definition of our in-flight services and a tool for mission validation. In the

current paper, we go beyond by implementing a complete UTM architecture. We integrate in-flight services to handle unexpected conflicts that may occur while the UAS are flying, and we showcase the performance of our system through heterogeneous use cases.

Finally, regarding the implementation of particular in-flight services, there are different approaches for conflict resolution and emergency management. Many works [28–30] have focused on flight planning and scheduling at a strategic level, i.e., in the pre-flight phase; though in-flight automated decision-making has not been properly covered in UTM systems. In general, given the massive search space to find optimal resolutions for conflicts in *Very Low-Level* (VLL) airspace scenarios, approximate solutions based on heuristic solvers [28] or lane maneuvers [30] predominate over optimal deconfliction approaches. In [31], a probabilistic framework is proposed to formulate the risk involved in UAS operations. That methodology could be integrated for automated, real-time data analysis in an emergency management solution. We take methodological ideas from these previous works, in order to implement conflict resolution and emergency management in our system considering the specifics of UAS operations in a civil airspace. However, the focus of this paper is more on the architecture design and integration, rather than on the particular algorithms for conflict resolution.

## 3. Design Framework

This section settles the framework for our UTM architecture. First, we analyze the desired properties and requirements for a UTM architecture from a design perspective. Then, we introduce ROS, which is the open-source middleware that we use to implement our architecture. We justify this selection by discussing the main features in ROS and how they fit our UTM system requirements.

### 3.1. Guidelines for System Design

The *Global UTM Association* (GUTMA) is a non-profit consortium of worldwide UTM stakeholders, and it has promoted a discussion about which key properties should be present in future UTM systems [13]. After reviewing their technical report, we came up with a summary of these key features for UTM systems. We believe that the following aspects should be taken into account during the design phase of any UTM architecture:

- **Digital**. The process of system digitization consists of making the communication between the different actors and components digital, and introducing automated decision-making procedures. This is a key aspect in UTM to reduce the operators' workload in an efficient and secure manner. Moreover, it enables the real-time exchange of data between the relevant parties for situation awareness and an easier integration of the UTM services.
- **Flexible and modular**. A UTM architecture should be flexible and adaptable to incorporate new actors (e.g., stakeholders) and functionalities (e.g., services), as they appear. Besides, the system should be modular, i.e., made of composable and reusable modules, in order to ease the process of creating more complex functionalities.
- **Scalable**. A scalable architecture is needed to grow with new actors and services. In order to achieve that, not only is the aforementioned modularity desirable, but also a paradigm with distributed responsibilities, rather than the obsolete scheme with a centralized ATC.
- **Safe and secure**. These two features are top priorities in any UTM ecosystem. In this sense, the system should know who is flying each unmanned aircraft, where they are flying (or intend to fly) to, and whether they are conforming (or not) to mandatory operating requirements.
- **Automated**. A UTM system providing automated services to assist the UAS operators will be more efficient and secure. Therefore, the system should provide support through automated functionalities for flight planning, monitoring, and real-time deconfliction, in order to ensure safe operations for both manned and unmanned aircraft.

- **Open-source**. The use of open-source technologies is preferable, as they offer a global approach towards creating and evolving the necessary services and protocols for scalable operations. Moreover, open-source components can speed up the development and the deployment of UTM services.

*3.2. Robot Operating System*

*Robot Operating System* (ROS) is an open-source framework for robot software development. It consists of a collection of libraries, tools, and conventions to ease the creation of complex applications in robot systems; including hardware abstraction, low-level device control, implementation of commonly-used functionalities, message-passing between processes, and package management. ROS is also well known among the UAS community, as it allows drivers to communicate with a wide spectrum of both open-source and commercial autopilots and onboard sensors. The use of ROS for multi-UAS systems is extending fast, as it paves the way for integration of heterogeneous hardware and software systems. ROS is a framework based on multiple processes (so-called *nodes*) that run in a distributed fashion. These processes can be grouped into *packages*, and communicate with each other by passing *messages*, which are typed data structures. On the one hand, ROS implements asynchronous communication through a publish/subscribe paradigm where nodes can stream messages over different *topics*. On the other hand, synchronous communication is implemented through *services* for request/response interactions.

We decided to use ROS as middleware for our UTM architecture because it offers multiple features that fit our design guidelines. First, ROS is designed to create modular and reusable components, and its preferred development model is to write ROS-agnostic libraries with clean functional interfaces. Therefore, ROS yields flexible and scalable systems that can be adapted easily to incorporate new functionalities. Second, ROS is open-source and strongly supported by a large community. Its federated system of code repositories enables collaboration and fast development for UAS complex systems. Communication solutions and drivers for most popular autopilots (e.g., PX4, ArduPilot, DJI, etc.) are already available in ROS. Moreover, ROS provides remarkable tools for system integration and testing, and there exist multiple options for multi-UAS simulation, including *Software-In-The-Loop* (SITL) solutions for common autopilots [32].

ROS also presents some issues for multi-UAS systems. Mainly, its centralized nature due to the existence of a single *master* node that handles all the procedures for node registration, and its lack of proper *Quality of Service* (QoS) policies. However, there exist efficient solutions for these issues. Multi-master architectures have already been used for applications with multiple UAS [6]; and the adoption of ROS 2 is growing fast among the community, with a smooth transition from primary ROS. ROS 2 proposes a fully distributed scheme, where each node has the capacity to discover other nodes, without the need for a central master. Since it is built on top of the industrial standards DDS (*Data Distribution Service*) and RTPS (*Real-Time Publish-Suscribe*), ROS 2 is capable of offering multiple QoS options for improved communication.

Even though we have chosen ROS to implement our UTM architecture, mainly due to its advantages for system integration and realistic SITL simulation, it is important to remark that the proposed UTM architecture is a more general concept, and it could be adapted to alternative middleware solutions.

**4. UTM System Architecture**

This section describes our proposed UTM system architecture. Figure 4 depicts an overview of all the software modules involved, as well as their interactions. The modules in green implement specific U-space services. As it was explained in Section 2.1, we focus on those services that are required to address unexpected events during the flight operation of a UAS. In particular, we cover four services with their corresponding modules: *Tracking*, *Monitoring*, *Emergency Management* (EM), and *Tactical Deconfliction* (TD). Besides, our system includes additional software modules that provide support to the UTM architecture. First,

there is a *Data Base* (DB) component that is in charge of handling all the relevant information about the state of the airspace, for instance, the current flight plans and tracks for all UAS operations (which are updated by the Tracking module) and the active geofences (which can be activated externally by auxiliary stakeholders like fire brigades or internally by the Emergency Management module). Second, the *U-space Service Manager* (USM) is a key module that acts as an interface between the UTM system and the rest of the U-space ecosystem. Basically, it receives state information and alerts from both the UAS and the external auxiliary stakeholders, and it communicates back recommended actions to deal with threatening events. These recommendations are generated by means of the interaction between the Tracking, Monitoring, Emergency Management, and Tactical Deconfliction modules.

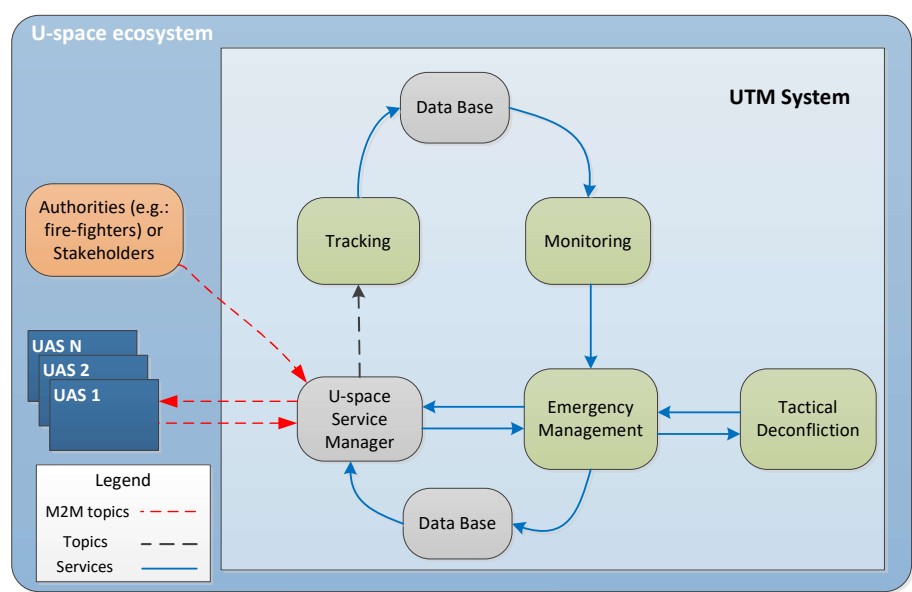

**Figure 4.** Overview of the proposed UTM system architecture. This system would be running in a remote server named U-space Service Provider Platform. The red arrows indicate remote communication links with other machines in the ecosystem.

Our system is built upon ROS (Section 3.2) and hence, each module consists of a software process implemented as a ROS node. The communication between modules takes place through ROS topics and services. In particular, the system is designed to use services in a preferable manner, as they provide the possibility of acknowledging message reception, which is crucial to reliably manage many of the UTM interactions. In those cases, one of the modules acts as a server while others act as clients, which results in an asynchronous communication between the modules. Upon a client request, the server module will carry out the requested activity and then it will reply, indicating whether the result was successful or not. Nevertheless, there are also a few cases where ROS topics are needed. Topics provide a synchronous communication, and they are used by modules that need to publish information at a constant rate.

In the following sections, we will provide a more detailed description of the different modules in our UTM system. For each module, we describe its functionality and interactions with other modules, as well as the methodology that we have used to implement them.

### 4.1. U-Space Service Manager

The U-space Service Manager is a key module in the UTM system, as it provides an interface with the rest of the actors in the U-space ecosystem, i.e., the UAS operators and auxiliary stakeholders like the airspace authorities, the fire-fighters or the police.

First, the USM receives positioning measurements from the control station of each UAS, which is transmitted by their onboard telemetry and ADS-B transceivers (if available). This information is forwarded to the Tracking module in order to keep updated a list of tracks for all the operational UAS. Second, the USM receives warning information that may be relevant for the UTM system, coming from external stakeholders (e.g., a declaration of a wildfire by the fire-fighters) or from the UAS (e.g., the detection of a jamming attack or a technical failure due to a lack of power). A jamming attack consists of an attempt to jeopardize the GNSS (*Global Navigation Satellite System*) signal of a UAS. These previous events are treated as possible threats by the system and are forwarded to the EM, which is in charge of processing them. Last, the USM communicates back to the UAS operators any action determined by the EM (e.g., an immediate landing or an alternative flight plan). Due to regulatory restrictions, the actions involving the variation of a UAS flight plan are just recommendations that must be confirmed or rejected by the corresponding pilot. In case of acceptance, the USM would notify the DB to update the state of that operation and its flight plan.

*4.2. Data Base*

The function of the Data Base module is to handle a digital data base with the required information to represent the situation of the current UAS operations, in the airspace managed by the UTM system. Basically, this information is made up of active geofences and UAS operations. The DB works as a server for the rest of the UTM system and hence, other modules can read the database in order to carry out their tasks (e.g., the Monitoring module uses the UAS predicted trajectories to detect events of lack of separation); or they can write the database to update the airspace situation (e.g., the USM can notify new accepted flight plans and the new EM geofences).

The DB manages two types of objects internally: geofences and UAS operations. Tables 2 and 3 depict the data structures for each of these objects. A geofence is a 4D portion of the airspace (a 3D geometrical space with an activation period of time) which has special restrictions for UAS, like flight prohibition. In the UTM context, the term *dynamic* geofence is used for those created during the UAS operation, while the *static* geofences are set in a pre-flight phase. The DB stores for each geofence in the airspace the following information: a unique identifier, its type (cylindrical or polygonal), its geometry definition, its minimum and maximum altitude, and its starting and finishing time instants. Besides, the DB stores each UAS operation, which consists of the following data: a unique identifier for the UAS, given by its ICAO (*International Civil Aviation Organization*) address; the priority level of the operation; its associated flight plan; the next waypoint assigned to the UAS; the predicted trajectory of the UAS; a brief description of the UAS operation; and the sizes of the Flight Geometry and the Operational Volume.

**Table 2.** Attributes of a geofence object.

| Attribute | Data Type | Description |
|---|---|---|
| Identifier | Integer | Unique number for geofence identification |
| Type | Enum | Cylindrical or polygonal |
| Geometry | List of 2D waypoints | Definition of the horizontal shape, defined by a circle or a polygon |
| Min/max altitude | Float | Altitude range where the geofence is active |
| Start/end time | Float | Time period in which the geofence is active |

**Table 3.** Attributes of a UAS operation object.

| Attribute | Data Type | Description |
|---|---|---|
| Identifier | Integer | Unique identification of the aircraft |
| Priority | Enum | Priority of the operation in the airspace |
| Flight plan | List of waypoints (x, y, z, t) | Reserved 4D trajectory for the operation |
| Next waypoint | Integer | Waypoint index that the UAS is currently targeting |
| Predicted trajectory | Float | Prediction of the future UAS trajectory |
| ConOps | String | Description of the concept of the operation |
| Flight Geometry | Float | Radius of the cylindrical volume where the UAS is intended to remain during its operation |
| Operational Volume | Float | Radius of the outer cylindrical volume to account for environmental or performance uncertainties |

### 4.3. Tracking

The Tracking module implements the U-space service with the same name. According to the U-space definition (Section 2.1), the main functionality of this service is to track the operational UAS in the airspace. These tracks contain information updated in real time about the UAS current position and its predicted trajectory within a certain time horizon. The module computes the tracks by fusing information from different sources that it receives through the USM. In particular, measurements from the UAS telemetry and ADS-B transceivers (when available) are integrated to achieve a more accurate estimation of the UAS positions. Moreover, the future trajectory of each UAS is predicted given its current position and velocity, as well as its flight plan. The tracking component keeps updated the UAS tracks in the DB module, so that this information is available for the rest of the system.

Mathematically, the Tracking module implements a stochastic filter that maintains a list of objects to estimate the state of each UAS, as depicted in Figure 5. This filter allows the system to cope with noisy and delayed measurements, as well as irregular sensor rates. The state of each UAS consists of its 3D position and velocity (expressed in *Universal Transverse Mercator* coordinates), and its current waypoint, i.e., the next waypoint of the flight plan that the UAS is targeting. The continuous variables are estimated through a *Kalman Filter* that integrates the measurements coming from the UAS telemetry and the onboard ADS-B transceivers. These data are previously transformed from geographic to Universal Transverse Mercator coordinates.

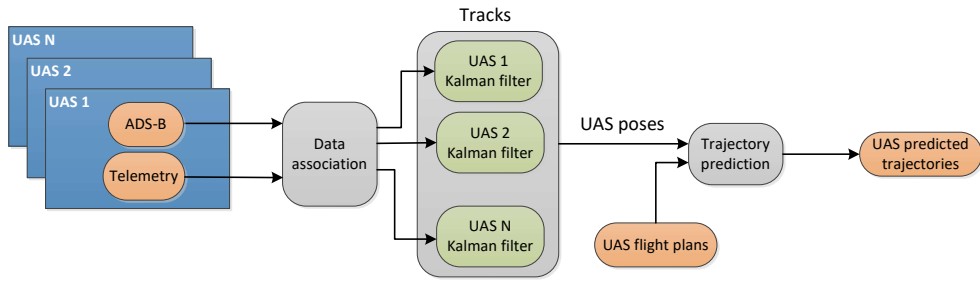

**Figure 5.** Scheme with the internal components of the Tracking module. The *data association* component matches the measurements from the UAS with their tracks, to update the corresponding Kalman filters. The future UAS trajectories are predicted using the tracks and the flight plans.

The procedures is as follows. At a constant rate, the list of operations is read from the DB in order to identify the active UAS. The state of all those UAS is predicted and then updated with the received observations. Each observation can be easily associated with its corresponding track, since they all come with a unique UAS identifier. The observations

with *unknown* identifiers are ignored by the filter, as they are considered non-cooperative aircraft. Moreover, the current waypoint for each UAS is computed by searching for the waypoint in its flight plan that is closest to its current position. The future trajectory within a given time horizon is also predicted for each track. If the current position of the UAS is close enough to its current waypoint (according to a given distance threshold), the prediction of the future trajectory sticks to the flight plan. Otherwise, the Kalman Filter is used to predict a trajectory given the current UAS position and velocity. Finally, after each step, the Tracking module updates all the information about the tracks in the DB module.

*4.4. Monitoring*

The functionality of the Monitoring module is to monitor the state of the airspace and to detect potential conflicts or threats that need to be managed by the UTM system. In particular, the module deals with conflicts related with UAS trajectories. Thus, it detects: (i) whether a UAS gets out of its reserved flight volume; (ii) whether it is in conflict with a geofence; or (iii) whether two UAS lose a minimum required separation. For that, the Monitoring module periodically reads information from the DB about the UAS tracks and the geofences, and it analyzes that information to determine when a threatening situation should be reported to the EM. When the Monitoring notifies the EM, it indicates the type of the detected threat, a prediction of the time instant when the event will occur and a snapshot with the current predicted trajectories of the involved UAS. This last piece of information is sent so that the modules resolving the conflicts use exactly the same data to evaluate the situation and hence, time glitches and incoherent solutions are avoided.

The first type of issue that is evaluated by the Monitoring module is related to the *Operational Volume* that is reserved by each UAS operation (see Figure 6). The Operational Volume is a 4D space that consists of a 3D volume around the flight plan with a temporal component representing the time that the volume, as part of an operation, will be reserved in the U-space ecosystem. The Operational Volume is composed by: the *Flight Geometry*, which defines the volume of airspace where the UAS is intended to remain during its operation; and the *Contingency Volume*, which is an outer surrounding volume to account for environmental or performance uncertainties. The closest distance between the current UAS position and its flight plan is computed to determine whether the UAS is out of its Operational Volume.

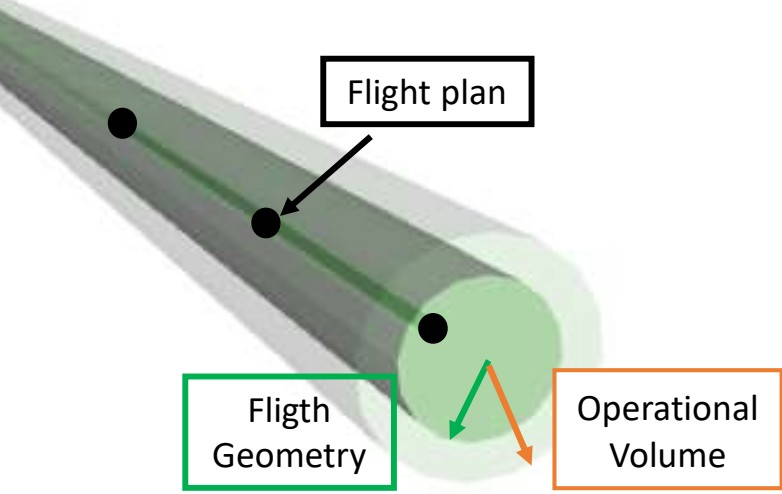

**Figure 6.** Graphical representation of the Operational Volume of a UAS operation (the orange arrow represents its radius). Given a flight plan, the green cylindrical volume around would represent its Flight Geometry (the green arrow indicates its radius), whereas the outer volume is the Contingency Volume.

In addition, this module monitors possible intrusions in geofences. For that, every waypoint belonging to the predicted trajectory of each UAS is compared against the active geofences, to determine whether the UAS is already intruding a geofence or it is estimated to enter one in a short future time. This check is carried out in 4D, i.e., the 3D volume of the geofence and its activation time are taken into account. More specifically, apart from checking the waypoint altitude with the minimum and maximum altitudes of the geofence, an evaluation on the horizontal plane is done depending on the shape of the geofence. If it is cylindrical, the distance of the given waypoint to the cylinder center is computed and compared with the geofence radius. If the geofence is defined by a polygonal shape, the *signed angle* method is applied. This method computes the sum of the angles between the segments that connect the test waypoint and each pair of points in the polygon. If this sum is $360°$, the waypoint is within the polygon, whereas it is outside if the sum is $0°$. Figure 7 depicts an example.

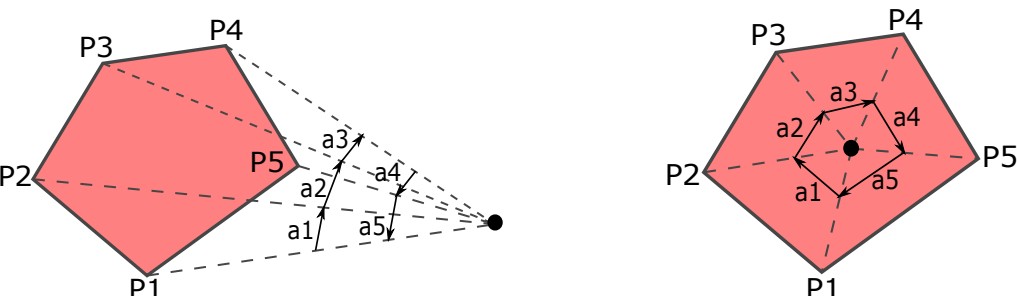

**Figure 7.** The signed angle method is used to evaluate whether a tested waypoint (black dot) is inside or outside a polygonal geofence. (**Left**), an example where the angles of an external waypoint sum up to $0°$. (**Right**), an interior waypoint whose angles sum up to $360°$.

Finally, the Monitoring module checks whether there is any loss of separation between UAS that needs to be notified. This check is done with a geometrical approach whose details can be seen in [33]. Basically, the idea is to discretize the airspace to model it as a 4D grid (see Figure 8), where each cell represents a 4D volume in space and time ($dX$, $dY$, $dZ$, $dT$) and stores a list of all the UAS whose trajectory is estimated to be inside. Thus, each waypoint of a UAS trajectory only needs to be compared with other waypoints within the neighboring cells (space and time neighborhood). For each waypoint in the 4D grid, the distances to the waypoints in the lists of its neighboring cells are calculated. If any of these distances is shorter than a safe distance, a threatening event of loss of separation will be reported.

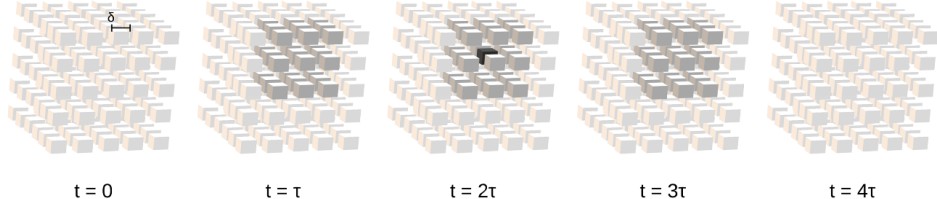

<div align="center">

t = 0     t = τ     t = 2τ     t = 3τ     t = 4τ

</div>

**Figure 8.** A 4D grid representation of the airspace. The dark grey cells would be the neighboring cells of the black cell.

### 4.5. Emergency Management

The Emergency Management module is the component of the UTM system that handles the threatening or unexpected situations in the U-space ecosystem. The module centralizes all the information related to the events that may become a threat, either due to conflicting UAS operations or to external warnings (e.g., a jamming attack or a bad weather situation). After analyzing the threatening events, the EM determines which are

the recommended actions to resolve the conflicts, and it sends them to the corresponding UAS operators.

The EM is a central module in the UTM architecture and, as such, it interacts with the Monitoring, the USM, the DB, and the TD. The possible threats or conflicts are notified to the EM by the Monitoring or the USM modules. The former reports about conflicts related with the UAS flight plans, as it was explained in Section 4.4. The latter reports about external warnings coming from UAS technical issues, UAS operators or auxiliaries stakeholders in the U-space. For instance, this is the case of a jamming attack, a bad weather forecast, the declaration of a wildfire by the fire brigades or any other threatening event notified by emergency corps.

Depending on the severity of each threat, the EM executes a decision-making procedure to determine the best possible actions to solve the conflict [34]. In this procedure, the EM takes into account the current flight plans for the involved UAS, the priority of their operations, and other restrictions in the airspace like the geofences. As output, the EM can decide to take three different types of actions: (i) to send a specific command to a particular UAS to terminate the flight, to go back to the flight plan, etc.; (ii) to create a geofence to isolate the detected threat; and (iii) to propose an alternative flight plan to one or several UAS to resolve the conflict.

In the first type of action, the EM acts, sending a notification to the corresponding UAS operator through the USM. In the second type of action, the EM creates a geofence and it interacts with the DB in order to update the database with geofences. In the third type of action, the EM sends the alternative recommended flight plans to the USM, which is in charge of forwarding them to the corresponding UAS. For the computation of these alternative plans, the EM receives the support of the TD module, which is requested to compute a series of alternative routes for the involved UAS, depending on the situation. The TD generates these routes by applying a set of predefined maneuvers for each UAS (see Section 4.6). Then, the EM selects which are the best alternative routes for all the UAS in conflict by minimizing the following value function:

$$\sum_{i=1}^{N} \sum_{j=1}^{M} \alpha \cdot c_{ij} + \beta \cdot r_{ij} \; ; \tag{1}$$

where $N$ and $M$ represent the number of conflicting UAS and the number of available maneuvers for each UAS, respectively; $c_{ij}$ is the cost incurred if the UAS $i$ executes the route $j$; $r_{ij}$ is the riskiness associated with the route $j$ executed by the UAS $i$; and $\alpha, \beta \in [0, 1]$ are the optimization weights. Each type of UAS maneuver considered by the TD will generate an alternative route for the UAS, with an associated cost and riskiness. The former is related to the additional time that the UAS has to travel to execute the route, while the latter measures the risk level of the route, e.g., how close it comes to other existing flight plans or geofences. The values of the weights assigned to the two terms need to be tuned by a human designer. In general, the system should favor safety over efficiency, so higher values for $\beta$ than for $\alpha$ are expected.

Finally, it is important to remark that all the actions sent by the EM to the UAS are just recommendations. According to the current regulation of the U-space ecosystem, the UTM can only suggest automatically possible correction actions, but those must be accepted or rejected by each UAS operator eventually. Nonetheless, our approach would be able to accommodate a UTM system where the whole process is executed autonomously without the need for human intervention, which is the final objective in the U-space framework.

### 4.6. Tactical Deconfliction

The Tactical Deconfliction module provides support to compute alternative flight plans for UAS that need to resolve a potentially threatening or conflicting situation. The TD receives requests from the EM indicating the necessary information related to the event to solve, i.e., the type of threatening situation and the data of the affected operations and

the active geofences. Depending on the situation, the TD will attempt different types of maneuvers to generate a list of alternative flight plans for the involved UAS. For each possible solution, the TD will compute an associated cost and riskiness level, which will be reported back to the EM, together with the generated alternative flight plans. Then, as it was explained in Section 4.5, the EM is the module that makes a final decision about which the best solution to resolve the conflict is.

The TD uses two different approaches to compute the alternative routes, depending on whether the threat is a conflict between different UAS or a situation with a single UAS involved. The first case occurs when the flight plans of several UAS are in conflict, e.g., due to a loss of separation. In that case, a geometric approach based on repulsive forces is used to modify the original flight plans. The details of the implemented algorithm can be seen in [35], but it basically models the UAS trajectories as cords with electrical charges that repel each other, in order to increase their separation. By applying vertical or horizontal separation maneuvers between the involved UAS trajectories in an iterative procedure (see Figure 9), the TD can generate several alternative solutions. The priorities of the conflicting flight plans are also considered. The algorithm tends not to modify the flight plans of those UAS whose operations present a higher priority in the U-space. For each computed solution, its cost is calculated as the total distance traveled by the UAS, whereas its riskiness is the length of the UAS routes that still get in conflict with other geofences. Even though these types of conflicts are solved in an iterative manner, by applying the tactical deconfliction procedure for each pair of UAS sequentially, the final solution could still produce additional conflicts with geofences. In this case, the Monitoring module would report those new pending conflicts in subsequent iterations.

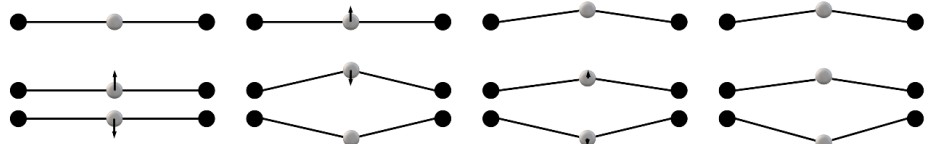

**Figure 9.** Iterative procedure to solve a conflict in the case of a loss of separation (from **left** to **right**). The flight plans of the two lower UAS are in conflict and need to be separated. Then, the middle UAS enters in conflict with the upper UAS, so these two get separated again. As the plan of the middle UAS gets modified, the lowest UAS is also adapted to achieve a final solution without loss of separation.

A second approach is used to solve situations with a single UAS involved. This is the case of a UAS that presents a technical problem, that is out of its Operational Volume, or that has a conflict with a geofence. In all those cases, a heuristic path planner based on the well-known A* algorithm is used. First, if the UAS flight plan goes through a geofence, the path planner generates an alternative route avoiding that geofence (see Figure 10, left). Second, if the UAS is already within a geofence, it gets out of the geofence through an *escape* point, and then it avoids the geofence to resume with its flight plan afterwards (see Figure 10, right). The TD also computes an alternative route from the current UAS position to the last waypoint in its flight plan, in order to skip the conflicting part of the plan and fly directly to the final goal. Third, if a UAS is out of its Operational Volume, two alternative routes are computed: one from the current UAS position to the closest point of its flight plan; and another from the current UAS position to its next waypoint in the flight plan, regardless of how long the UAS remains out of its Operational Volume.

In the three cases, an alternative route to return back to the home station is also computed. The EM could select this option if all the other solutions to continue with the operation are too risky. In all the generated solutions, the cost is determined by the total distance traveled by the UAS. The riskiness is determined by the minimum distance between the alternative route and any geofence, or by the length of the route portions that remain within a geofence, in case that the solution goes through any geofence partially. In

case of a UAS out of its Operational Volume, the riskiness of the solution is determined by the length of the route portion where the UAS stays out of its Operational Volume.

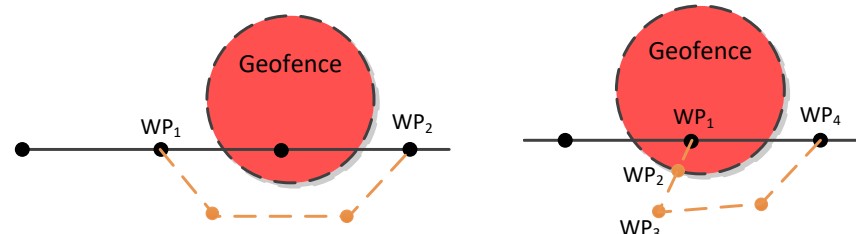

**Figure 10.** (**Left**), a UAS with a flight plan crossing a geofence. The last waypoint of its flight plan before entering the geofence ($WP_1$) and the first waypoint after leaving it ($WP_2$) are obtained, and this portion of the flight plan is replaced by an alternative route (dashed line). (**Right**), a UAS that is inside a geofence. The escape point ($WP_2$) is that on the geofence's border closest to the UAS ($WP_1$). From $WP_3$, which is already at a safety distance from the geofence, to the first point of the flight plan after leaving the geofence ($WP_4$), an alternative route avoiding the geofence is inserted to modify the original flight plan.

Finally, an alternative route where the UAS travels to its closest landing site can also be computed in some cases, for instance, if the UAS presents a technical problem like a lack of battery. In those cases, the riskiness is determined by the distance of the route that goes through any geofence in the airspace.

### 4.7. Discussion

In this section, we discuss the functionalities implemented by the U-space services of our UTM architecture, when compared to those expected in the current definition of the U-space ecosystem. For that, we have summarized in Table 4 the expected functionalities to be covered by each of the U-space services included in our system, according to the bibliography studied in Section 2.1. In the following, we discuss which capabilities are already covered by our system and the missing points for future implementations.

**Table 4.** Summary of the functionalities to be covered by the U-space services included in our UTM system.

| U-Space Service | Functionalities | Covered in Our UTM System |
|---|---|---|
| Tracking | Cooperative UAS tracking | ✓ |
| | Non-cooperative UAS tracking | ✗ |
| | Capability to exchange data with other services | ✓ |
| | Real-time tracking with data fusion from multiple sources | ✓ |
| | Tracking data recording | ✓ |
| Monitoring | Air situation monitoring | ✓ |
| | Non-cooperative UAS identification | ✗ |
| | Flight non-conformance detection | ✓ |
| | Restricted area infringement detection | ✓ |
| | Provision of traffic information for UAS operators | ✗ |
| | Conflict alerts | ✓ |
| Emergency Management | Emergency alerts | ✓ |
| | Provision of assistance information for UAS operators | ✓ |
| Tactical Deconfliction | Transmission of deconfliction information from the USM to the UAS | ✗ |
| | Transmission of deconfliction information in real time | ✓ |

- *Tracking*. This service is supposed to consider cooperative and non-cooperative UAS, but our current implementation only manages cooperative UAS. This is because we have focused on enabling automated decision-making for the operating UAS, which makes no sense for non-cooperative vehicles. Those should be treated as uncontrollable intruders (i.e., threats) in the airspace. However, our Tracking module does have the capability to update and record data in real time from different sources. Other services can also access these data through the DB module if needed.
- *Monitoring*. As in the previous case, our current implementation does not consider non-cooperative UAS. We did not establish a specific communication link to provide traffic information to the UAS operators either, though this could be easily done through the USM. However, our Monitoring module does accomplish all the other expected functionalities, i.e., it detects and alerts in real time about conflicts related to flight non-conformances, geofences, and inter-UAS separation.
- *Emergency Management*. This service is expected to provide the UAS operators with notifications about alerts and any other emergency assistance. Besides, our EM module includes automated decision-making capabilities, in order to manage threats in real time by proposing safe and optimal actions to the UAS.
- *Tactical Deconfliction*. Although this service is supposed to provide deconfliction information to the UAS operators through the USM, in our scheme this role is played by the EM module. This is because the automated decision-making capability is implemented in the EM module, which uses the TD module to get support generating possible alternative plans. Then, the EM is the one in charge of deciding the best option for real-time deconfliction.

## 5. Experiments

This section contains experimental results to showcase the capabilities of the proposed UTM system. The objectives of these experiments are twofold: (i) we show the integration of the complete architecture, with all its functional modules interacting together to accomplish the specified UAS operations; and (ii) we demonstrate our system operating in real time in a realistic setup, to test its capabilities to solve different types of conflicts in an automated manner. For that, we have defined two use cases (Section 5.1) involving heterogeneous UAS and several types of conflicts, in order to validate all the modules in our UTM system. The tested use cases are realistic both in terms of the UAS operational parameters and the experimental setup (Section 5.2). Our experiments were carried out by means of *Hardware-In-The-Loop* (HITL) simulations where the UAS operators and the UTM framework ran at different physical locations, with a real long-distance communication link in between. All of the results of the tests are described in Section 5.3.

### 5.1. Use Cases Definition

We defined two use cases using the heterogeneous UAS that were depicted in Figure 2: the multirotor DJI M600 and the fixed-wing Atlantic I. These UAS are used in the GAUSS project to run tests integrating aircraft with different maneuverability and different proprietary autopilots. Both use cases involve a pair of UAS performing operations with different or equal priorities, and both require the interaction of all the modules of the proposed UTM system.

Figure 11 depicts a top view of each use case, with the corresponding initial flight plans. Table 5 summarizes the operational parameters for the use case 1. $UAS_1$ is a multirotor performing an operation for precision agriculture, while $UAS_2$ is a fixed-wing aircraft that has to inspect an electrical power line. Given its easier maneuverability, the priority of the $UAS_1$ operation is set lower. The initial flight plans (see Figure 11, left) are such that the UAS do not coincide in space and time throughout their operations. However, we simulated an unexpected delay in the start of the $UAS_1$ operation, which resulted in a later violation of the minimum safety distance between both UAS. Thus, this use case is used to test how the UTM is able to detect a loss of separation between the UAS and to perform

real-time tactical deconfliction for an inter-vehicle conflict, deciding new flight plans for both UAS.

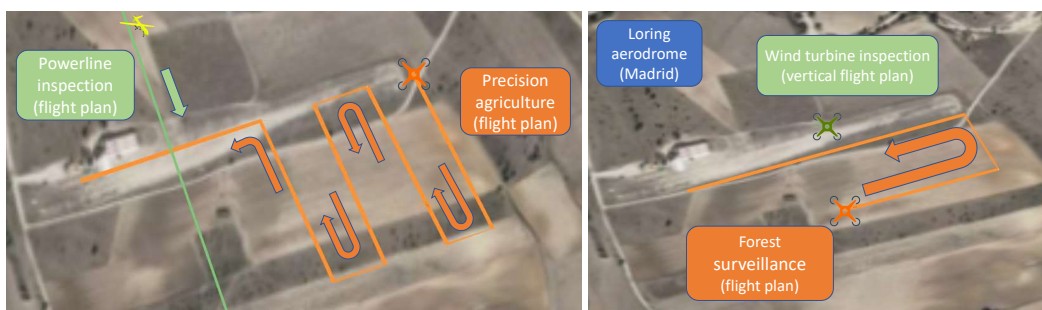

**Figure 11.** Top views including the initial flight plans of the use case 1 (**left**) and the use case 2 (**right**). All the operations were planned in an area of the Loring aerodrome in Madrid (Spain).

**Table 5.** Operational parameters for the use case 1.

|  | **Operation 1.1** | **Operation 1.2** |
| --- | --- | --- |
| ConOps | Precision agriculture | Powerline inspection |
| UAS type | M600 (UAS$_1$) | Atlantic I (UAS$_2$) |
| Cruising speed | 3.3 m/s | 30 m/s |
| Altitude (Above Ground Level) | 40 m | 100 m |
| Operation priority | Low | High |
| Events involved | Loss of separation | Loss of separation |

Table 6 summarizes the operational parameters for the use case 2. In this case, both UAS$_1$ and UAS$_2$ are multi-rotors, performing two operations with equal priority. In their initial flight plans (see Figure 11, right), UAS$_1$ moves on a vertical line to accomplish the inspection of a wind turbine, while UAS$_2$ has to fly on a horizontal plane to survey a nearby forest. During the operation, a jamming attack is simulated over UAS$_1$. The objective of this use case is to test how the UTM is able to react in an automated manner to an emergency generated by an external source, creating a new geofence and adapting to the conflicting flight plans.

**Table 6.** Operational parameters for the use case 2.

|  | **Operation 2.1** | **Operation 2.2** |
| --- | --- | --- |
| ConOps | Wind turbine inspection | Forest surveillance |
| UAS type | M600 (UAS$_1$) | M600 (UAS$_2$) |
| Cruising speed | 1 m/s | 1 m/s |
| Altitude (Above Ground Level) | 30–90 m | 70 m |
| Operation priority | High | High |
| Events involved | Jamming attack | Geofence conflict |

### 5.2. Experimental Setup

We have developed our UTM system architecture in ROS (Kinetic version), and the software is available online (https://github.com/grvcTeam/gauss, accessed on 26 April 2021). First, we used an airspace SITL simulation based on Gazebo [27] for system integration and preliminary tests. Then, we setup a realistic environment to run experiments in real

time with HITL simulations. These experiments were carried out within the framework of the GAUSS project, with the configuration depicted in Figure 12.

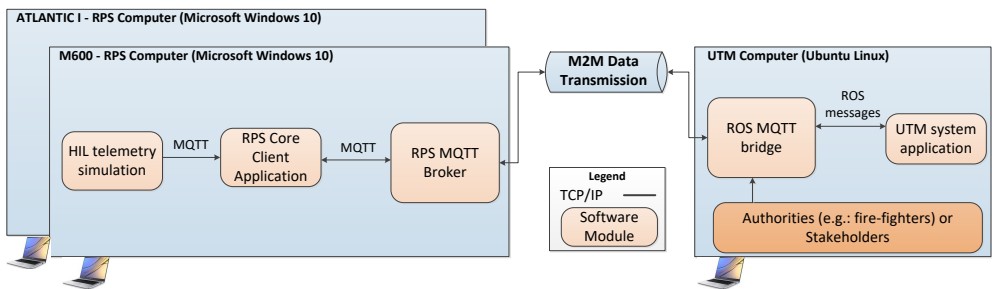

**Figure 12.** Setup for the experiments. The computers running the RPS for the two UAS and the UTM system were placed at remote locations and communicated through the Internet via the MQTT protocol.

The company EVERIS (https://www.everis.com/global/en, accessed on 26 April 2021) ran on its headquarters in Madrid (Spain) a *Remote Pilot Station* (RPS) for each type of UAS. Each RPS has an integrated HITL simulation producing real-time telemetry data for the operating UAS, a graphical user interface to show this telemetry, and the operational information to the safety pilot (*RPS Client Application*), and an *RPS MQTT Broker* to communicate data over the Internet. The RPS Client Application was developed by the company SATWAYS (https://www.satways.net, accessed on 26 April 2021) and it can be seen in Figure 13). Simultaneously, we ran our UTM system on a server located in Seville (Spain), connected to the Internet via a *ROS MQTT bridge*. The UAS RPS communicated with the remote UTM system exchanging JSON (*JavaScript Object Notation*) messages sent over the MQTT (*Message Queuing Telemetry Transport*) transport protocol (We used the open-source Apache Active MQ broker). Moreover, the time synchronization for the exchanged data between the remote computers was achieved thanks to an NTP (*Network Time Protocol*) server. It is important to highlight that this experimental setup is close to the real U-space ecosystem, where the UTM system would be running on a server located at a remote distance of the UAS operators.

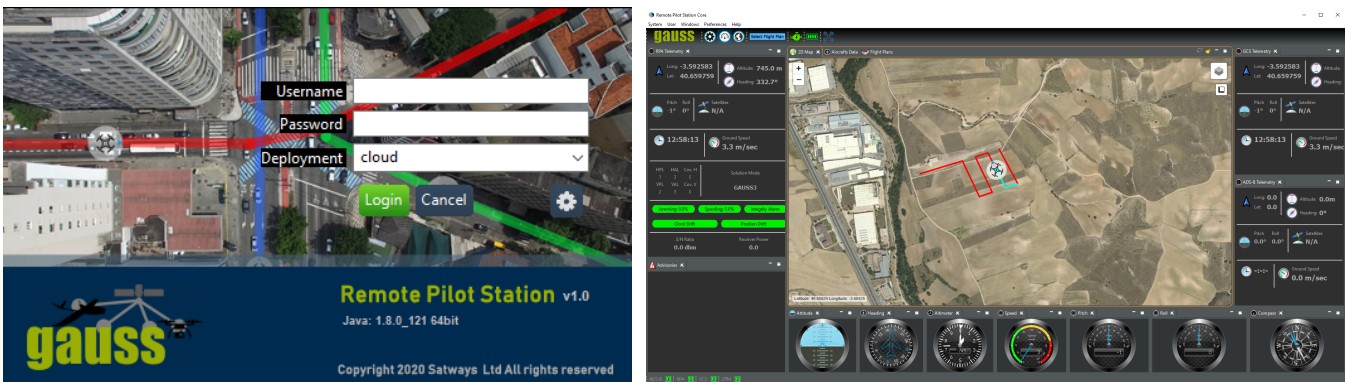

**Figure 13.** Screenshot of the graphical user interface developed by SATWAYS running on the RPS Client Application.

*5.3. Results*

In this section, we present results of the experimental tests for the two proposed use cases (an illustrative video with the use cases can be seen at https://grvc.us.es/downloads/videos/UTM_System.mp4, accessed on 26 April 2021), with all the modules in our UTM system working together. It is important to highlight that the experiments were carried out in real time, with the UTM system monitoring the operations and managing the unexpected

events properly. Moreover, the proposed solutions to solve the conflicts were executed in an automated manner by the simulated UAS, and supervised by human safety pilots.

Figure 14 shows a timeline for the experiment of the use case 1. Both UAS were supposed to start their operations simultaneously ($t = 0$ s) according to their pre-flight generated plans, without conflicts. However, we simulated a delay of 3 s in the start of the $UAS_1$ operation. The Tracking module received periodically positioning information from both UAS and it updated the DB accordingly. The Monitoring module checked for conflicts periodically using the updated tracks from the DB and, at $t = 24$ s, it detected a future loss of separation conflict between the UAS. This was communicated to the EM, which ran an automated decision-making process (supported by the TD) to propose the optimal conflict resolution. In this case, an alternative flight plan was sent to $UAS_1$ through the USM module. Figure 15 shows the initial flight plans for the UAS and their reserved Operational Volume. Despite not having conflicts initially, the delay in the $UAS_1$ operation provoked an eventual loss of separation in the last part of its operation, which was resolved with an alternative flight plan. Figure 16 depicts the three options generated by the TD module and the optimal solution (in terms of risk and traveled distance) selected by the EM. In the experiment, the conflict was detected by the UTM system well in advance, and the total time between the detection and the communication of a solution to the USM took 0.13 s.

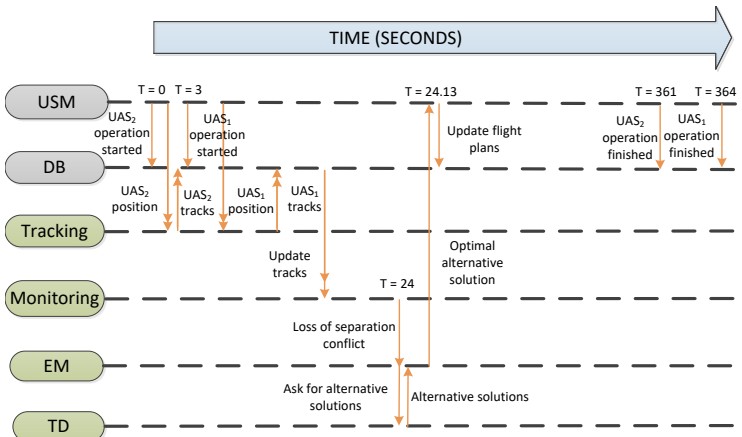

**Figure 14.** Timeline of the experiment of the use case 1, where a loss of separation event is resolved. Single arrows indicate isolated interactions between modules, whereas double arrows indicate periodic communication.

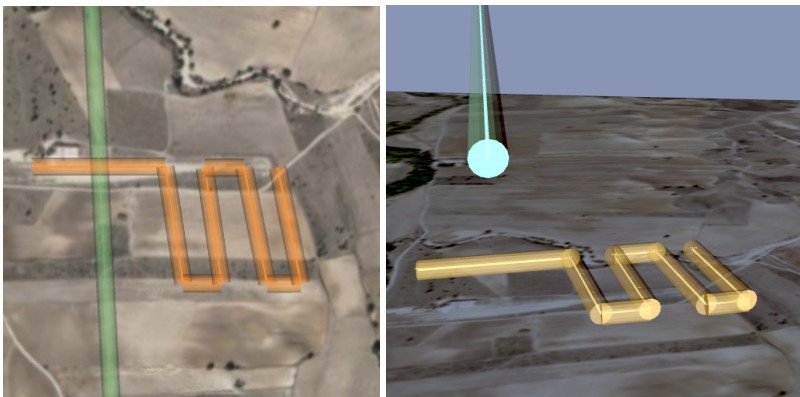

**Figure 15.** A top (**left**) and a perspective view (**right**) of the initial flight plans in the use case 1. The Operational Volumes are shown for both UAS. There are no conflicts given the UAS 4D trajectories.

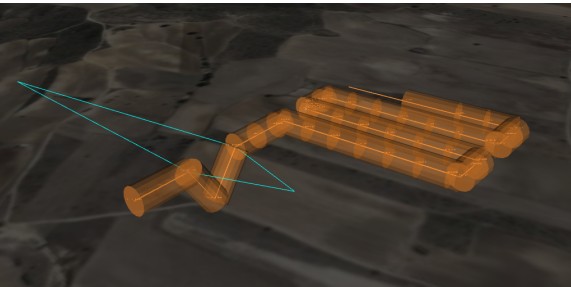

**Figure 16.** A perspective view of the conflict resolution in the use case 1. A new flight plan for $UAS_1$ (with a final *go down* maneuver) was selected to keep the safety distance with $UAS_2$. The other alternative maneuvers generated by the TD module (*go left* and *go right*) are also shown.

Figure 17 shows a timeline for the experiment of the use case 2. Both UAS started their operations simultaneously ($t = 0$ s) following pre-flight plans without conflicts. The Tracking module received periodically positioning information from both UAS, and it updated the DB accordingly. The Monitoring module checked for conflicts periodically using the updated tracks from the DB. We simulated a jamming attack over $UAS_1$ ($t = 12$ s) that was notified by the USM to the EM, which ran an automated decision-making process. In this type of threat, due to the involved risks, the EM decided to suspend the $UAS_1$ operation (notifying the USM) and to create a geofence around (updating the DB). Then, the Monitoring module detected ($t = 15$ s) a future geofence conflict with the $UAS_2$ flight plan, which was resolved by the EM (with the support of the TD) with an alternative plan avoiding the geofence. Again, the time between the detection of the conflict and the communication of the optimal solution to the USM was less than 1 s. Figure 18 shows the initial flight plans for the UAS and their reserved Operational Volumes, and the situation right after the jamming attack. Despite not having conflicts initially, the creation of a new geofence provoked an eventual conflict, which was resolved with an alternative flight plan for $UAS_2$ (see Figure 19).

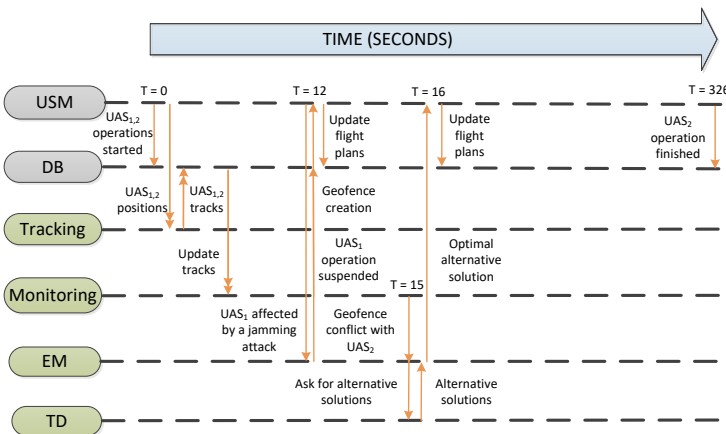

**Figure 17.** Timeline of the use case 2, where a jamming attack and a geofence conflict are resolved. Single arrows indicate isolated interactions between modules, whereas double arrows indicate periodic communication.

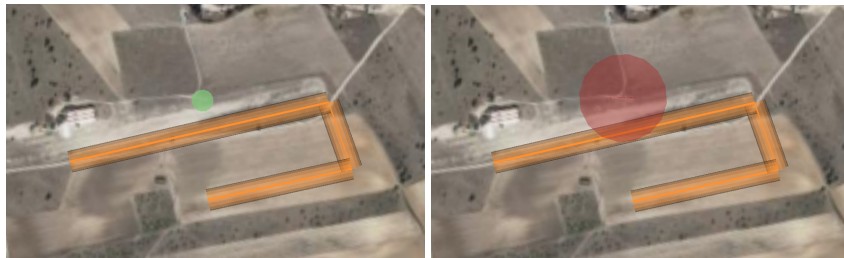

**Figure 18.** (**Left**), top view with the initial flight plans in the use case 2. The Operational Volumes without conflicts are also shown. (**Right**), situation after the detection of the jamming attack. A geofence (in red) is created around the attacked UAS, which generates a conflict with the flight plan of the other UAS.

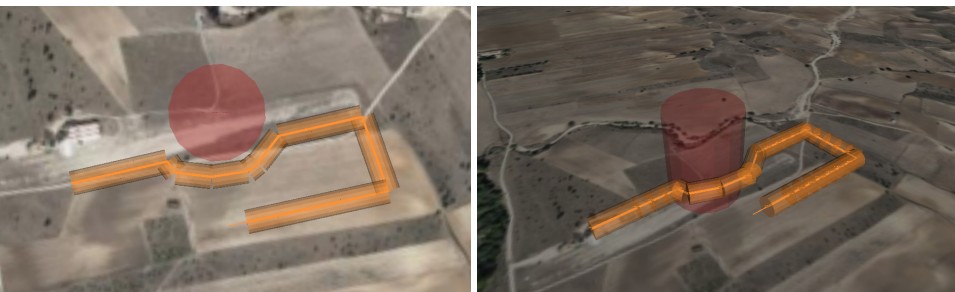

**Figure 19.** A top (**left**) and a perspective (**right**) view of the optimal solution in the use case 2. An alternative flight plan for UAS$_2$ is generated to avoid the geofence.

Finally, it is important to recall that the experiments were carried out with a setup where the UTM system ran at a remote distance of the UAS stations. Despite that, the communication delays and response times by the UTM system were adequate for a real-time resolution of the unexpected conflicts. In particular, we measured a reception of the UAS telemetry data at the USM of an average rate of 1 Hz with a maximum delay of 40 ms.

## 6. Conclusions

In this paper, we have presented a UTM system architecture framed within the U-space ecosystem. Our software architecture is flexible and general, and it is built as an open-source solution that could be easily extended with additional U-space functionalities. Nonetheless, we have focused on in-flight services for automated threat management and conflict resolution, which is a major gap in the current state of the art. In our realistic experimental setup, with the involved systems running HITL simulations communicated through a remote link with the UTM system, we have demonstrated that the proposed UTM solution is capable of managing unexpected events in real time, proposing solutions in an automated manner. In our experiments, the system was able to detect and resolve different types of conflicts, reasoning about 4D UAS trajectories and Operational Volumes. Besides, we have tested the feasibility of the system for the future U-space, integrating heterogeneous types of UAS (fixed and rotary wing), heterogeneous positioning technologies (ADS-B and telemetry from different autopilots), and a database to keep track in real time of the different UAS operations and geofences.

Our system has still some practical limitations. First, it relies on a centralized UTM server that requires continuous communication with the other actors. This bottleneck could be addressed by splitting the UTM system into a set of distributed and interconnected servers. Second, our approach does not consider non-cooperative vehicles in the VLL airspace, such as ultralight planes, nor pre-flight services. However, the architecture is flexible enough to integrate additional services, e.g., for flight operation pre-planning. Besides, non-cooperative vehicles could be tackled by working with see&avoid systems on board the UAS. As future work, we plan to develop further on the emergency management functionality, analyzing the possible threats that could appear in the VLL airspace,

and quantifying the involved risks of the alternative solutions proposed by our system. Thus, we will be able to improve the capabilities of the system to solve more conflicts safely and efficiently, and to test it in more varied use cases. Furthermore, we plan to adapt our UTM system to ROS 2 and to validate it in field trials within the framework of the GAUSS project, which will be a significant step toward a totally automated U-space environment.

**Author Contributions:** Conceptualization, C.C., Á.C. and J.C.; software and validation, C.C. and H.P.-L.; writing and editing, C.C., H.P.-L. and J.C.; supervision, Á.C., J.C and A.O.; funding acquisition, A.O. All authors have read and agreed to the published version of the manuscript.

**Funding:** This work has received funding from the European Union's Horizon 2020 Research and Innovation Programme under grant agreement No 776293 (GAUSS).

**Institutional Review Board Statement:** Not applicable.

**Informed Consent Statement:** Not applicable.

**Data Availability Statement:** The original contributions presented in the study are included in the article; further inquiries can be directed to the corresponding author.

**Conflicts of Interest:** The authors declare no conflict of interest.

## abbreviations

The following abbreviations are used in this manuscript:

| | |
|---|---|
| ADS-B | Automatic Dependent Surveillance Broadcast |
| ATC | Air Traffic Control |
| ATM | Air Traffic Management |
| DB | Data Base |
| DDS | Data Distribution Service |
| EASA | European Aviation Safety Agency |
| EM | Emergency Management |
| GNSS | Global Navigation Satellite System |
| GUTMA | Global UTM Association |
| HITL | Hardware-In-The-Loop |
| ICAO | International Civil Aviation Organization |
| JSON | JavaScript Object Notation |
| MQTT | Message Queuing Telemetry Transport |
| NASA | National Aeronautics and Space Administration |
| NTP | Network Time Protocol |
| QoS | Quality of Service |
| ROS | Robot Operating System |
| RPS | Remote Pilot Station |
| RTPS | Real-Time Publish-Subscribe |
| SITL | Software-In-The-Loop |
| TD | Tactical Deconfliction |
| UAS | Unmanned Aircraft System |
| USM | U-space Service Manager |
| UTM | Unmanned aerial system Traffic Management |
| VLL | Very Low Level |

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
