# Peer review of "Unmanned Aerial Traffic Management System Architecture for U-Space In-Flight Services"

_applsci, doi:10.3390/app11093995_

Round 1

Reviewer 1 Report

Comments to the Author

A review report on Unmanned Aerial Traffic Management System Architecture for U-space In-Flight Services (Manuscript number: applsci-1184101)

The reviewer suggests a minor revision to the manuscript of applsci-1184101.

  1. To address the contribution of this paper to the readers, please try to state the major insights and suggestions instead of stating “…we performed our tests with the UTM system and the operators of the aerial aircraft located at remote locations…”
  2. The reviewer suggests that the authors should state clearly the contribution, managerial insights, theoretical and practical implications, limitations, and future research in the concluding session.
  3. Section 4, Figure 4. The authors proposed a UTM system architecture. But, the reviewer suggests that the authors should state the rationale for providing a cloud-based “cyber-layer” system architecture without connecting to the “physical layer”. The reviewer suggests re-organizing Figure 4 and Figure 12 to ensure a well-developed architecture considering the data transmission.

Author Response

We would like to thank the reviewer for his/her time devoted to our paper and the given feedback. In the following, we provide replies to the reviewer's comments:

1. The sentence that the reviewer is referring to is located in the abstract. What we meant is that we placed different parts of the system at remote locations in order to set a realistic experimental setup, with the communication issues that would be involved in a real scenario. Despite that, our system was able to handle events in real time. We have modified the abstract to reflect better that idea.

2. We have extended the last part of the conclusion section to include a discussion about the main practical limitations of our architecture and how they could be addressed.     

3. We agree with the reviewer that maybe the term "cloud" server was confusing. In our architecture, the UTM system is running on a remote server, but a cloud-based architecture using cloud services is not mandatory. We used the word "cloud" to indicate that the UTM was run in a remote place, different than where the UAS operators are located. However, we have decided to replace the term "cloud" by "remote" server, as we think this help clarifying the architecture. Also, we have modified Figures 4 and 12 to indicate clearly which connections imply machine-to-machine physical data transmission.  

Reviewer 2 Report

In this paper, the authors developed an open source UTM architecture for in-flight services for automated conflict resolution and threat management. Overall, the authors have done a good job explaining their system and I commend them on their use of visualizations.

I have the following minor suggestions.

  1. Table 1 lists five services that are covered in UTM system, but the UTM system architecture described in Section 4 (page 7) only covers four, excluding the tactical geofencing service.
  2. The future of “aircraft” is “aircraft” without the “s” at the end.
  3. The paper overuses sentence conjunctions, like “thus” and “besides,” which are not always used correctly. They take away from the message and sometimes misinterpret the message.

Author Response

We would like to thank the reviewer for his/her time devoted to our paper and the given feedback. In the following, we provide replies to the reviewer's comments:

1. According to [1], Tactical Geofencing is defined as as service that is in charge of sending messages to the drone user/operator to report about geofences. This task is implicitly implemented in our UTM system by the U-space Service Provider module, and this is why we marked it as covered although there is no a specific software component to implement it. However, in order to avoid any confusion, we have decided to unchecked the Tactical Geofencing service in Table I.

2.    We have revised the manuscript to replace the term "aircrafts" with "aircraft".

3.    We have revised the whole manuscript to improve the use of conjunction words, avoiding them when unnecessary. 

REFERENCES
[1] CORUS project, “U-space Concept of Operations,” no. October 2019, pp. 1–92, 2020.